# Catalytic Ring-Opening Polymerisation of Cyclic Ethylene Carbonate: Importance of Elementary Steps for Determining Polymer Properties Revealed via DFT-MTD Simulations Validated Using Kinetic Measurements

**DOI:** 10.3390/polym16010136

**Published:** 2023-12-31

**Authors:** Daniel Brüggemann, Martin R. Machat, Reinhard Schomäcker, Mojgan Heshmat

**Affiliations:** 1Institut für Chemie—Technische Chemie, Technische Universität Berlin, Straße des 17. Juni 124, D-10623 Berlin, Germanyschomaecker@tu-berlin.de (R.S.); 2Covestro Deutschland AG, Kaiser-Wilhelm-Alle 60, D-51373 Leverkusen, Germany; 3Institute of Technical and Macromolecular Chemistry, CAT Catalytic Center, RWTH Aachen Universität, Worringerweg 2, D-52074 Aachen, Germany

**Keywords:** DFT-metadynamics, kinetics, ring-opening polymerisation, cyclic ethylene carbonate, biodegradable polymers, aliphatic cyclic carbonates

## Abstract

The production of CO_2_-containing polymers is still very demanding in terms of controlling the synthesis of products with pre-defined CO_2_ content and molecular weight. An elegant way of synthesising these polymers is via CO_2_-containing building blocks, such as cyclic ethylene carbonate (cEC), via catalytic ring-opening polymerisation. However, to date, the mechanism of this reaction and control parameters have not been elucidated. In this work, using DFT-metadynamics simulations for exploiting the potential of the polymerisation process, we aim to shed more light on the mechanisms of the interaction between catalysts (in particular, the catalysts K_3_VO_4_, K_3_PO_4_, and Na_2_SnO_3_) and the cEC monomer in the propagation step of the polymeric chain and the occurring CO_2_ release. Confirming the simulation results via subsequent kinetics measurements indicates that, depending on the catalyst’s characteristics, it can be attached reversibly to the polymeric chain during polymerisation, resulting in a defined lifetime of the activated polymer chain. The second anionic oxygen of the catalyst can promote the catalyst’s transfer to another electrophilic cEC monomer, terminating the growth of the first chain and initiating the propagation of the new polymer chain. This transfer reaction is an essential step in controlling the molecular weight of the products.

## 1. Introduction

Increasing concern about climate change, connected to the use of (diminishing) fossil resources, necessitates the development of new, sustainable technologies from renewable sources for the generation of chemicals, energy, and materials [1,2,3,4,5]. To this end, cyclic ethylene carbonate (cEC in short), due to ecological and economic reasons, and its ready availability from ethylene oxide and carbon dioxide, seems to be a very attractive monomer and reagent for polyether carbonate production [6,7]. The obtained aliphatic polycarbonates are of interest due to their new and valuable applications in biodegradable and biocompatible materials. Hence, the presence of carbonate functional groups in the polymer chain facilitates the depolymerisation of aliphatic polycarbonates. Interestingly, in ring-opening polymerisation (ROP) of cEC, the polymers formed may have lower densities than the monomers (volume expansion may accompany polymerisation), which can be beneficial for industrial applications [8,9,10,11]. The catalysed ROP of cEC is a low-pressure technology, representing a technical advantage compared to the direct copolymerisation of ethylene oxide (EO) and CO_2_ [12,13,14,15,16,17,18,19]. Both thermodynamic and kinetic factors are crucial for the ROP of cEC. It is worth noting that the maximum temperature for the homopolymerisation of CEC is below 25 °C, but polymerisations have also occurred at over 100 °C [17]. At high polymerisation temperatures, the resultant polymers’ repeat units are a mixture of monomeric units (carbonate units) and the corresponding oxide units, meaning that CO_2_ release occurs during the polymerisation. The release of the CO_2_ makes the overall reaction ΔS positive, so this polymerisation becomes thermodynamically possible only at high temperatures. In cEC polymerisation, the purpose is not to completely avoid CO_2_ release, but keeping CO_2_ to some extent in the final polymer product is advantageous [17]. To overcome the kinetic stability of cEC and polymerise the cEC, modern catalysts and catalytic processes have been developed in the last few decades to achieve the ROP quickly. Notably, an efficient catalyst must conduct the reaction in such a way that CO_2_ release is minimised during the polymerisation of cEC. A series of recently investigated catalysts approved for cEC polymerisation is of particular industrial interest, i.e., they are structurally simple, commercially available, and efficient catalysts. The relatively high conversion values and CO_2_ content percentages that were experimentally obtained and approved by the Covestro company [4,5,20] in the ROP of cEC confirm that these catalysts are exciting candidates with the potential for obtaining a more fundamental understanding with kinetic studies and computational methods. Among the examined series of 30 different catalysts, we selected K_3_VO_4_, K_3_PO_4_, and Na_2_SnO_3_ as representative examples for the current study. The cyclic carbonate conversion reactions cover carbonate formation and ether linkages in the final product, along with some CO_2_ release (Figure 1).

For the selected catalysts, the cations are either equal (K_3_PO_4_ and K_3_VO_4_) or, if they are different, no significant effect from the cationic counterpart is observed in simulations and experiments (Na_2_SnO_3_ vs. K_2_SnO_3_). Hence, the main difference in catalytic performance for the selected catalysts is caused by the anionic part, and we have considered the anionic pathway (Figure 2) for the current study. Anionic ROP (Figure 2) begins with a nucleophilic attack by an anionic initiator, such as the anionic counterpart of the catalyst, or can be generated, e.g., via proton abstraction from starter alcohol. The nucleophile can either attack the carbonyl carbon or the methylene carbon of the cyclic carbonate [21,22,23].

In this work, we aim to understand the details of the interaction between the catalysts and cEC monomer, employing DFT-MTD simulations and kinetic measurements to shed more light on the elementary steps for the CO_2_ release mechanism and differences between various considered catalysts. The main questions that we address are the consumption of the entire starter molecule in the initiation step, which pathways are dominant in the propagation step throughout the interaction between catalyst and monomer, and the possible effects on the CO_2_ content and molecular weight of the polymer are.

Recent findings obtained using advanced ab initio molecular dynamics (AIMD) simulations showed that the possibility of identification of alternative pathways of various complex reactions could be challenged in a dynamic picture [24,25,26,27,28,29]. This motivated us to investigate the ROP of cEC in the presence of the catalysts, as mentioned above, using DFT-metadynamics simulations as the computational method of choice. We employ metadynamics simulations to calculate the free energy surfaces (FESs) for the ROP and CO_2_ release mechanisms at a defined temperature. Simulation of a free energy landscape rather than a zero Kelvin potential energy surface allows the incorporation of entropic contributions due to the flexibility of motion of free molecules in the reaction environment. According to previous studies, the entropic contributions affect the formation of molecular complexes from small molecules and their respective transition state barriers of formation [30,31,32,33].

Understanding the underlying mechanisms and identifying the critical steps of catalysed ring opening and chain growth of cyclic carbonates and CO_2_ release pathways is the basis for computational chemistry-assisted research, leading to improved catalyst structures and performance. Using the results and insights obtained in this work, novel catalyst candidates can be rationally designed for better performance and process optimisation.

To make this possible, comparing the computer-calculated values with experimental ones is an elegant approach. By determining the kinetics, it is possible to understand the reaction better. Understanding the kinetics also opens up the control possibilities of ring-opening polymerisation. The mechanism of the ROP is not fully clarified, so it is still unclear why the basicity has an influence and why specific catalysts no longer react during the reaction [6]. These questions are, therefore, the central focus of this paper, and with them, the clarification of the mechanism of ring-opening polymerisation.

These polymers give access to a wide-ranging product portfolio. For example, using a hydrophobic initiator, functional non-ionic surfactants can be produced that are both environmentally friendly due to their excellent biodegradability and can be recycled. For example, these surfactants have already been proven to purify the microplastic contamination of water [34]. The production of low-molecular polymers (700–1200 g/mol) for the production of surfactants was the aim of the funded project on which this work is based. However, the findings of this work can also be applied directly to higher degrees of polymerisation.

## 2. Materials and Methods

### 2.1. Computational Details

All DFT calculations, including geometry optimisations and ab initio molecular dynamics (AIMD) simulations, were performed using the CP2K program [35] with the Gaussian and plane-wave (GPW) method. The valence orbitals were expanded in the DZVP-MOLOPT Gaussian basis set in combination with Goedecker, Teter, and Hutter pseudopotentials and were used with a plane wave cutoff energy of 280 Ry. We used the PBE density functional [36] augmented with Grimme D3 dispersion correction [37]. The criterion for self-consistent field convergence was set to 5.0 × 10^−7^. The AIMD simulations were done in the NVT ensemble, with the temperature controlled by a CSVR (canonical sampling through velocity rescaling) thermostat, set at various temperatures (413, 423, 433, and 443 K), and a period of 500 fs. The MD time step was 0.5 fs. To investigate the reaction mechanism at finite temperature, characterise the reaction pathway, and identify the transition state region between reactant and product states, we performed metadynamics simulations using the PLUMED 2.8 plugin in combination with CP2K [38,39,40]. The simulations were initiated from the optimised molecular structures and conducted until several transition state re-crossing events could be observed. To prevent sampling from unnecessary regions of the FES, harmonic walls with force constant K = 250 kcal/molÅ^2^ were used for CVs. The Gaussian bias potentials were added every 100 steps. Multiple test simulations were run to set the computational parameters, including the number and type of CVs, the height and width of Gaussian bias potentials, and quadratic walls for each part of the reaction mechanism.

We performed our simulations on a molecular cluster schematically shown in Figure 3. As depicted in Figure 3, the molecular cluster consists of six cEC monomers, one alcohol molecule (F = 1 for simplification), and a catalyst molecule. The typical catalyst molecule includes the anionic (MO_4_^3−^ or MO_3_^2−^) and cationic (A^+^) counterions.

We considered six cEC monomers in our simulations since six cECs can make the first shell interact with the catalyst with a reasonable computational cost for AIMD calculations. In fact, we investigated the interaction between the catalyst anion and cEC monomer, and a significant number of molecules participating in reaction mechanisms is one of each species. To estimate the interactions between cEC monomers, ROH starter, and catalyst–anion, we have probed the variation of average H-bond distances between ROH and six surrounding cEC monomers (ROH⋯O(cEC)) in the first solvation shell through an unbiased equilibrium MD simulation in Appendix A. The ROH-cEC distances are compared with the ROH-O(anion-CAT). The main interactions involved are those between ROH and the anion (shorter distances) that result in proton transfer in the initiation step of polymerisation. The interaction between ROH and O-cEC (according to distances) is weaker and O-cEC mainly interacts with the surrounding cations. Hence, ROH does not exist that long in the reaction medium and is deprotonated by the anion of the catalyst. After the proton abstraction, the main H bond is between the protonated catalyst anion and the surrounding molecules (cEC monomers and alkoxide), which can cover the protonated anion sufficiently. The reaction temperature in the experiment is above 150 °C, and the catalysts are expected to be solubilised entirely under these conditions. Additionally, lab results indicate that the catalyst cannot be separated adequately via filtration after the reaction. Both facts conclude that considering homogenous reaction conditions is a valid starting point. The collective variables (CVs) were the relevant distances in each reaction step. We used VMD 1.9.4 software to visualise the MD trajectories and snapshots [41].

### 2.2. Chemicals Used and Suppliers

The following chemicals with the indicated purities were used for the experiments. Ethylene carbonate (cEC, 99%) was obtained from Alfa Aesar (Haverhill, MA, USA). Ethylene glycol (99%) was obtained from Roth (Karlsruhe, Germany). Methanol (99.9%) was obtained from VWR (Dresden, Germany). Dichlormethan-D2 (99.5%) was obtained from Roth (Karlsruhe, Germany). Tichlormethan-D1 (99.8%) was obtained from Roth (Karlsruhe, Germany). Potassium phosphate tribasic anhydrous (99.5%) was obtained from VWR (Dresden, Germany). Sodium orthovanadate (V) (99.9%) was obtained from Alfa Aesar (Haverhill, MA, USA) and Potassium tin (IV) oxide trihydrate (95%) was obtained from Alfa Aesar (Haverhill, MA, USA).

### 2.3. Experimental Method

All reactions proceeded at the same conditions using a 50 mL reactor with a heatable jacket. The reactor has a GL25 opening for measurement with an in situ IR probe and two feed valves for operation under inert gas. The reactor was heated with a Haake F6 thermostat. Silicone oil AP200 was chosen as the operating fluid for the thermostat. The IKA RCT Basic magnetic stirrer was used for mixing. A Mettler Toledo Reakt IR 15 was used for the in situ measurements. The spectrometer has a Si probe connected to the spectrometer via AgX 9.5 mm × 1.5 mm fibre optics. The measured wavenumber range is from 4000 cm^−1^ to 800 cm^−1^ with a resolution of 4 cm^−1^. The reactor was brought to experimental temperature according to Appendix A, and the in situ IR probe was supplied with fresh nitrogen (l) for cooling the detector. The blank value of the IR probe was measured with the empty reactor, in which only the magnetic stirring bar was present. After measuring the blank spectrum, cyclic ethylene carbonate (20 g, 1 eq.), which had been preheated in a 60 °C water bath, was added to the reactor, the stirrer was set to 600 RPM, and the in situ measurement was started At least 10 min waiting time was used to obtain a baseline measurement for cEC. Ethylene glycol (10 mol%, 0.1 eq.) and catalyst (1 mol%, 0.01 eq.) were weighed beforehand and added quickly to the reactor. As the reaction progressed, the changes in the characteristic bands were monitored in the Software IC IR 7 (see Figure 1): 1800–1760 cm^−1^ [42] for cEC (C-O-C plugin vibrations, strained system) [42], 1750–1700 cm^−1^ [42] for CO_2_ (linear unstrained system) [42] and the ether band at 1260 cm^−1^ [42], (these are the expected signals for a nascent poly(ethylene ether carbonate)) [42,43,44,45]. The reaction was allowed to proceed until the band typical for cEC at 1800–1750 cm^−1^ was diminished entirely. 

## 3. Results and Discussion

Our previous study considered the ring-opening polymerisation of cEC in the presence of the starter ROH, which begins with the proton abstraction from the starter ROH molecule by the catalyst anion (step I [46]). The generated RO^−^ attacks the C=O group of the first neighbouring cEC. Then, the second generated alkoxide attacks the next cEC (step II). This mechanism is shown in Figure 2, considering our model cluster. The rate-determining step for this mechanism is the cEC ring opening (step II). The values of the ΔG^≠^ for the two steps are reported in Table 1.

As shown in Table 1, the barriers are low in the presence of ROH (initiation). The barriers, shown in Table 1 for step II, are correlated to the second cEC opening through a nucleophilic attack by the first cleaved cEC. This step is the rate-determining step since the proton abstraction (step I) has the lower barrier. However, considering that the concentration of the starter molecule is less than the cEC monomer at a particular stage in the reaction, the starter is consumed, and only cEC and catalyst are present in the reaction environment. Therefore, the reaction proceeds throughout the catalyst–cEC interaction (propagation), and investigating this pathway is the primary purpose of the current study.

In this work, we have considered Na_2_SnO_3_, K_3_VO_4_, and K_3_PO_4_ as the catalysts for the cEC ring-opening polymerisation. From a previous study, we found that the M-O (central atom-oxygen) bond length has a crucial impact on the catalyst basicity, i.e., a longer M-O bond leads to more basicity (easier proton abstraction) but less nucleophilicity for a nucleophilic attack to the CH_2_ of the cEC ring. By the basicity, we refer to the basicity of the transition metal/main element oxides (K_3_PO_4_, K_3_VO_4_, Na_2_SnO_3_), inherently weaker bases than KOH and NaOH. The order of decreasing the M-O bond distance of the considered catalysts is Na_2_SnO_3_ > K_3_VO_4_ > K_3_PO_4_. These catalysts’ average M-O bond lengths are 1.939, 1.734, and 1.571 (in Å). The average O⋯cations distances are 2.271, 2.617, and 2.604 (in Å) for Na_2_SnO_3_, K_3_VO_4_, and K_3_PO_4_, respectively. The differences in the molecular structures of the catalysts severely affect their catalytic performance. Hence, a longer M-O bond distance performs better in keeping a higher CO_2_ content in the product. The planar structure of the SnO_3_^2−^ catalyst anion causes complexation of the alkoxide anion (RO^−^ generated from ROH) to the Sn centre for a while during the initiation step. This complexation delays the reaction in case of SnO_3_^2−^ vs. VO_4_^3−^, 9.5 h vs. 5.0 h, respectively, in experiment, which is in accordance with the higher barrier of the cEC ring opening for SnO_3_^2−^ vs. VO_4_^3−^ (Table 2). For a detailed analysis we refer to the [46]. In addition, the cation’s lability also plays a role in more efficient catalytic performance. Especially for the smaller anions such as PO_4_^3−^, bigger cations perform better than smaller ones. The ΔG^≠^ values of the ring opening decrease from H^+^ to K^+^ for phosphate anion, and in the experiment, the CO_2_ contribution increases with larger cations. This effect is due to the longer anion…ation distance and less-covalent character of the anion-cation interaction, leading to more accessibility of the catalyst anion in the reaction. For bigger anions such as SnO_3_^2−^ and VO_4_^3−^, switching between Na^+^ and K^+^ does not affect the catalyst performance, and due to the stronger ionic character, the ΔG^≠^ values and CO_2_-contribution do not change with changing Na^+^ to K^+^. The complete results of the cation influence on the ΔG^≠^ and the experimental CO_2_ contribution are reported in Table 2. We note that for H_3_PO_4_ and Li_3_PO_4_ catalysts in Table 2 only minor conversions (due to the reduced activity) were observed. However, the conversion was higher than 80 percent for the rest of the catalysts.

Figure 3 depicts the considered CO_2_ release mechanism triggered by the interaction between the catalyst anion and the CH_2_ of the cEC ring.

The examination of the CO_2_ release path of Figure 3 with MTD simulations confirmed that the higher nucleophilicity of the phosphate anion is the leading cause for easier CO_2_ release during the polymerisation of cEC. Considering that, the mechanism shown in Figure 3, which results in the attachment of the catalyst to the cEC ring, can proceed in parallel with the pathway shown in Figure 2. This work analyses this pathway and the elementary steps to conduct this molecular complex (between the catalyst anion and the cleaved cEC) to further chain growth and polymerisation. The experimental observations show that higher catalyst concentration increases the CO_2_ release. Additionally, at the beginning of polymerisation, the primary cause of the CO_2_ release is due to the cEC decomposition. Hence, the hidden elementary steps in the pathway shown in Figure 3 can affect the molecular weight of the polymer as well as the CO_2_ content and the overall reaction rate. We investigated this pathway in the current study to better understand the catalyst–cEC interaction and to address the role of catalyst characteristics.

### 3.1. Simulation of the Catalyst–cEC Attachment Path

To run metadynamics simulations on the path shown in Figure 3, we have considered the distance between the nucleophilic oxygen atom of the catalyst anion and the methylene carbon in the cEC monomer, i.e., O^−^⋯C(CH_2_-O), as the first collective variable (CV1) and the C(H_2_)⋯O(ethereal) inside the cEC as the second collective variable (CV2). The MTD simulations have been performed at 413, 423, 433, and 443 K. For brevity, we only show the results of 423 K; for the other temperatures, we include the data in Appendix A. We have shown the variation of the CV1 versus CV2 in Figure 4 for the trajectory of cEC cleavage and catalyst attachment at 423 K, in which the two reaction coordinates (two distances) are plotted versus each other (x, y axis for CV1 and CV2, respectively). Investigation of the patterns created by the CVs in each plot indicates some similarities and differences between the path of catalyst attachment for different catalysts that can lead us to different elementary steps. We note that the three catalysts’ computational parameters and temperatures are equal. As shown in Figure 4, for Na_2_SnO_3_ and K_3_VO_4_ catalysts, the catalyst attachment path goes through an intermediate region of CV1 ≥ 2.5 Å and CV2 ≥ 2.5 Å. On the other hand, in the case of K_3_PO_4_, a diagonal transition path (that directly connects the reactant state to the catalyst–cEC complex) with a more significant density of the transient states, which is invisible for the SnO_3_^2−^ and VO_4_^3−^, appears in the plot. Variation of CV1 versus CV2 in Figure 4 identifies different regions based on the density of the dots (transient structures). These regions are singled out in Figure 4 based on the values of the CVs. As can be seen in Figure 5 (left side), three stable states are observed: the reactant state (CV1 ≥ 2.5 Å, CV2 ≤ 1.6 Å), the intermediate state (CV1 ≥ 2.5 Å, CV2 ≥ 2.5 Å), and the state in which the CAT-cEC bond (complex) is formed (CV1 ≤ 1.6 Å, CV2 ≥ 2.5). The regions between these states indicate the transition state areas, i.e., TS^vert.^, TS^hor.^, and TS^Nu−attack^. On the right side of Figure 5, the schematic molecular structures corresponding to the highlighted regions are depicted.

In the case of Na_2_SnO_3_ and K_3_VO_4_, the mechanism begins with ring cleavage (ethereal CH_2_-O bond cleavage and intermediate formation) by passing a vertical transition state region (TS^vert.^). The intermediate either returns to the closed cEC or releases CO_2_. In the case of K_3_PO_4_, an additional path can be identified: the cEC cleavage and nucleophilic attack by the catalyst anion happen simultaneously, and a diagonal transition state region is observed (TS^Nu−attack^). These two distinguishable patterns can be identified in the calculated FESs according to the relative energies in kcal/mol. Figure 6 shows the calculated FESs at 423 K for the three catalysts. As seen in Figure 6, the stepwise path, which is the transition through the intermediate, can be observed in the pattern of the FES of Na_2_SnO_3_ and K_3_VO_4_. The FES of the K_3_PO_4_ identifies a new concerted path in addition to the stepwise, which corresponds to the diagonal transition in that the two CVs are involved simultaneously. The numbers at the FESs show the energies in kcal/mol. The more stable intermediate, in the case of Na_2_SnO_3_ and K_3_VO_4_, can conduct the reaction in both directions; that is, turning into the catalyst attachment and CO_2_-release or into the backward direction and generating the cEC, which involves an alkoxide attack and ring opening (Figure 2 Step II). In the case of K_3_PO_4_, a pattern with direct catalyst attachment (diagonal with simultaneous involvement of CVs) can be observed.

In summary, the different patterns observed for stannate/vanadate vs. phosphate reflect the different natures of the chemical interactions between the catalyst and cEC (non-similar catalyst characteristics). The more-nucleophilic character of the catalyst anion results in the sampling of a concerted pathway of the catalyst attachment. The following section compares the calculated results with model parameters determined from kinetic measurements to address the correlations/differences between simulations and experiments.

### 3.2. Experimental Analysis of the Kinetics of Polymerisation of cEC, including CO_2_ Release

The experimental investigations on the kinetics of the polymerisation of cEC indicate that higher CO_2_ content can be obtained by the stannate and vanadate anions, which confirms the better performance of these catalysts for the ROP of cEC. Notably, the kinetic measurements confirmed that, at the beginning of the reaction, the primary source of the CO_2_ release is the cEC cleavage (decomposition). The schematic presentation of the experimentally observed reaction network for SnO_3_^2−^ and VO_4_^3−^ anions is shown in Figure 4 and for PO_4_^3−^ in Figure 5. k_1_ and k_2_ in Figure 4 represent the rate constants of ring-opening polymerisation of cEC and the chain propagation with CO_2_ release, respectively, and k_3_ and k_4_ show the polymeric chain (PECn) decay due to decomposition or CO_2_ release from the chain, respectively.

The kinetics of these reactions can be described with first-order rate Equations (1)–(4).
(1)$$r_{1}=k_{1}\cdot c_{cEC}^{n}$$
(2)$$r_{2}=k_{2}\cdot c_{cEC}^{n}$$
(3)$$r_{3}=k_{3}\cdot c_{PECn}^{n}$$
(4)$$r_{4}=k_{4}\cdot c_{PECn}^{n}$$

The material balances of the involved components were combined from these rate equations and solved numerically. Using a Runge–Kutta algorithm, the rate constants were determined via fitting the model to the experimental data. For fitting the absorption curves, the concentrations of the corresponding species were multiplied with the extinction coefficients of these species, estimated from single component spectra (Figure 7 and Table 3). (For a detailed description of the parameter fitting, see Appendix A. For representative results of fitting the kinetic model to experimental data for all experiments at different temperatures, see Appendix A).

Table 3 reports the experimentally measured rate constants corresponding to the reaction network in Figure 4 (for SnO_3_^2−^ anion as the catalyst) at various temperatures. As seen in Table 3, k_3_ and k_4_ are negligibly low. Hence, the primary source of the CO_2_ release is a result of the interaction between monomer and catalyst (CO_2_ release from cEC). This result agrees with the mechanism shown in Figure 3, in which the CO_2_ release is due to the catalyst attachment. In Table 3, we also show the ratio of k_2_/k_1_ to compare the rate of CO_2_ release versus polymerisation of cEC. The relative values show that CO_2_ release occurs either slightly slower than polymerisation (entries 1–3) or at a similar rate to polymerisation (entry 4). The k_2_/k_1_ ratio is not strongly dependent on the temperature. This experimental observation agrees well with the computationally observed mechanism in Figure 4 and Figure 5 (for stannate and vanadate anions). Hence, the possibility of proceeding via catalyst attachment followed by the CO_2_ release or returning to the closed cEC and further involvement in the main polymerisation path (propagation via alkoxide attack) is identified.

Furthermore, the calculated FESs in Figure 6 show similar values of the ΔG^≠^ for converting the intermediate to the CAT-CH_2_ bond and forming the reactant state (18.5 vs. 18.7 kcal/mol, respectively). According to the experimental measurements, a similar trend of the k_1_ and k_2_ rate constant ratios can be observed for VO_4_^3−^, which implies similar catalytic characteristics of the VO_4_^3−^ and SnO_3_^2−^ anions. Additionally, as shown in Figure 6, in the middle part, for the backward path, i.e., returning to the reactant state from the intermediate state, the ΔG^≠^ is more favoured (i.e., 9.2 kcal/mol from intermediate to the reactant state). The results of kinetic measurements of VO_4_^3−^ and the rate constants are reported in Appendix A.

For the PO_4_^3−^ anion, the kinetic measurement results differ from those with SnO_3_^2−^ and VO_4_^3−^ as catalysts. The experimentally observed network of the reactions catalysed by PO_4_^3−^ is shown in Figure 5. In this Scheme, k_des_ indicates the rate constant of catalyst deactivation, which is observed for PO_4_^3−^. The results of the kinetic measurements and the rate constants are reported in Table 4. The literature (also shown in Table 4) shows that the ROP of cEC catalysed by phosphate has lower CO_2_ content in the final polymer (10% lower than K_3_VO_4_ and Na_2_SnO_3_) [46]. In addition to that, according to ^31^P-NMR, insertion of phosphate is observed during the polymerisation (^31^P NMR (500 MHz, CD_2_Cl_2_) δ 17.84 (s, 1P), 1.64 (d, J = 169.7 Hz, 7P), 1.08 (s, 2P)), shown in Figure 8. The peaks at 0–2 ppm can be assigned to potassium phosphate [47]. However, there is a signal at 17.84 ppm, which does not correspond to the signals of potassium phosphate. Instead, signals related to phosphonates can be found in this range [47].

In all experiments, below 170 °C, no complete reaction could be observed. Even adding fresh potassium phosphate did not lead to any further reaction. Only above 170 °C is it possible to complete the reaction, since the polymerisation reaction is faster than the deactivation of the catalyst. However, this leads to lower CO_2_ contents of the products. The determination of the conversion and the CO_2_ content was performed via the ^1^H-NMR of the products dissolved in CD_2_Cl_2_ or CDCl_3_ and the measurement of ^31^P-NMR in CD_2_Cl_2_ (see Figure 8). Similar to the case of stannate, k_3_, and k_4_ are also negligible for phosphate catalysis.

As Table 4 shows, the ratio of k_2_/k_1_ is very different from what was observed with stannate. For the first entry, it is close to one, which means the propagation reaction with CO_2_ release and the ring-opening polymerisation have similar rates. However, for entry two, the ring-opening polymerisation is faster than the propagation with CO_2_ release. This difference can also be seen from the high CO_2_ content. However, since the reaction did not proceed to full conversion and achieved only 7% yield, this measurement is highly susceptible to experimental error. For entries 3 and 4, the rate of the reaction with CO_2_ release is faster than cEC ring-opening polymerisation, producing a polymer with much lower CO_2_ contents. Due to the simultaneous deactivation of the phosphate catalyst, the evaluation of the kinetic parameter is greatly hampered, therefore resulting in large error bars of the deduced values.

Furthermore, the values of k_des_ versus k_1_ and k_2_ are considered. In that case, a closer correlation between the propagation with CO_2_ release (k_2_) and catalyst deactivation can be found, e.g., for entry four at 425 K (k_2_ and k_des_ are very close). In comparison to the computational results, this is in line with the concerted mechanism in the case of phosphate where the catalyst attachment to the cleaved cEC appears faster than in the case of stannate, due to a lower barrier (10 vs. 18.5 kcal/mol, respectively, Figure 6). This situation provides conditions for a faster CO_2_ release which is correlated to the lower barrier in simulated FES of K_3_PO_4_ for the diagonal path.

Considering the theoretical predictions and experimental results we have shown so far, the main question that still needs to be addressed is how the polymerisation proceeds through the generated molecular complex, in which the catalyst is bound to the monomer via the CH_2_ moiety (Figure 4 and Figure 5), and what is the probability of CO_2_ release alongside the chain propagation path starting from the catalyst–monomer molecular complex.

### 3.3. Alternative Pathways of Chain Growth Revealed by Simulations

One of the possible chain growth paths starts with the catalyst–anion attached to the monomer is shown in Figure 9. We have examined this path using MTD simulations with the considered CVs depicted in Figure 9. The considered CVs correspond to the COO^−^ nucleophilic attack and cEC cleavage (CV1 and CV2, respectively).

The analysis of the corresponding MTD trajectories for this path at four temperatures for the three catalysts indicates that, since the COO^−^ terminus has a relatively weak nucleophilic character than the RO^−^ (due to the negative charge delocalisation on the two oxygen atoms), the barrier for the nucleophilic attack to the second cEC increases. On the other hand, since the ΔG^≠^ for the CH_2_-O(ethereal) bond cleavage is, on average, 8–11 kcal/mol for various molecular systems and temperatures, the possibility of CO_2_ release upon cEC cleavage enhances before COO^−^ attachment. Alongside examination of this pathway, the possibility of 30% random CO_2_ release for some of the trajectories was observed (four trajectories ended with CO_2_ release out of 12 trajectories). Thus, to prohibit CO_2_ liberation and obtain the MTD trajectory for calculating the ΔG^≠^ values, we had to fix the corresponding bond lengths leading to CO_2_ liberation. These bonds, responsible for CO_2_ release and kept fixed during simulations, are highlighted in red in Figure 9. The calculated barriers for this reaction (ΔG^≠^) are shown in Table 5 for the rate-determining step, which is the nucleophilic attack by COO^−^ to the second cEC monomer (CH_2_ moiety).

As seen from Table 5, generally, the barriers decrease with increasing temperature, and for K_3_PO_4_, the barriers are lowest. However, chain propagation can proceed via this pathway without the starter ROH, with a higher barrier and more probability of CO_2_ liberation than the path including the starter ROH. This finding agrees with the experimental measurements that the majority of CO_2_ liberation is caused by cEC decomposition at the beginning of the reaction. In all paths, one reaction coordinate (CV2) is responsible for the ring opening of cEC to convert it to a nucleophile (COO^−^) that can further proceed with the ring opening of the next cEC (propagation).

As an alternative possibility, the insertion of the cEC monomer to the O(CAT)-CH_2_(cEC) bond, which is already formed in the attached catalyst–monomer complex, is examined. With the insertion paths, we estimate if a cEC can perform as a nucleophilic species to cleave the O(CAT)-CH_2_(cEC) bond. The insertion paths are shown in Figure 10. In the insertion paths **I** and **II**, we try to cleave the cEC ring and convert it to a nucleophile. However, due to the inherent weak nucleophilicity of cEC by itself, CO_2_ release alongside the insertion of cEC into the O(CAT)-CH_2_ bond is observed with a predominant possibility (in 80% of the trajectories).

This leads us to the conclusion that cEC is inherently a more electrophilic species than nucleophilic, especially when it interacts with the cations in the surrounding environment that pre-activate the cEC ring. These observations introduce an alternative pathway to the previously proposed mechanism, in which cEC attacks as a nucleophile to a second cEC (Figure 6) [6].

In insertion path **III**, the S_N_2 mechanism via the nucleophilic attack by the O(CAT) to the CH_2_ moiety of a neighbouring cEC is examined. We have run the insertion paths for all three catalysts at four temperatures and, after analysing all trajectories, besides CO_2_ release, two new pathways are revealed based on the MTD trajectories. We note that these new revealed paths were unknown. In one observed path, the second O of the PO_4_^3−^ anion attacks the CH_2_ of the second cEC, which means that chain growth can occur through a new channel via the second cEC cleavage. In the second observed path, the catalyst detachment for SnO_3_^2−^ and VO_4_^3−^ occurred. The two new observed paths are shown in Figure 11A,B. Path A was mainly observed for phosphate and path B for stannate and vanadate catalysts. The variation of the CVs versus the time evolution is depicted in Figure 12 for the trajectories with the two alternative paths. In addition to the CVs, the variation of the non-CV distances that lead to the new-unknown paths in Figure 11 is also depicted in Figure 12. As can be seen in Figure 12, alternative bond formation/dissociation can be probed in these trajectories instead of the considered CVs. Figure 12 upper part shows the second O(CAT)-CH_2_ bond is formed as soon as the cEC ring is cleaved (blue line). As shown for vanadate and stannate, two new bonds between the O(CAT)⋯CH_2_^+^ and O^−^(cEC)⋯central atom (V or Sn) are formed when the ring is cleaved (blue and orange lines, respectively). Observation of the paths shown in Figure 11 motivated us to examine the possibility of these methods of polymerisation and for the chain transfer (catalyst detachment) for the three catalysts throughout the calculation of the corresponding MD trajectories and ΔG^≠^ values.

To calculate the ΔG^≠^ for the two observed paths, we considered the appropriate CVs and simulated the MTD trajectories at 423 K. These CVs are shown in Figure 13, and the calculated ΔG^≠^ values are reported in Table 6. The ΔG^≠^ values correlate to the rate-determining step, corresponding to the nucleophilic attack by the O^−^ (CV1) in each pathway. In the chain transfer/catalyst-detachment path in Figure 13, we performed the transfer of a partially formed chain with an external nucleophilic attack (RO^−^). As seen in Table 6, the catalyst’s second oxygen attack has a lower barrier in the case of K_3_PO_4_ than K_3_VO_4_ and Na_2_SnO_3_, in agreement with the stronger nucleophilicity of the phosphate anion.

The chain transfer through the catalyst detachment has a high barrier, which indicates that the generated chain is firmly attached to the catalyst. Nevertheless, chain transfer occurs at a specific time during the growth period of each individual chain.

Since the barriers of the catalyst’s second oxygen attack are lower than the catalyst detachment (Table 6), the second nucleophilic attack begins faster and can promote the chain transfer/catalyst detachment. This finding means that with the formation of the second cleaved cEC and the start of a new polymerisation channel, the older generated chain leaves the catalyst, leading to polymer chains with specific shorter lengths and constant molecular weights.

These insights obtained using calculations are well in line with the experimental observations and results. The calculated rate constants of the chain transfer and catalyst-detachment (k_trans_), according to the kinetic measurements in the experiment, are reported in Table 7 compared to the rate constants of the ring-opening polymerisation and the propagation with CO_2_ release (Table 3 and Table 4). As seen in Table 7, the k_trans_ (chain transfer rate constant) is approximately an order of magnitude smaller than the k_1_ and k_2_ (the rate constants of polymerisation). This observation agrees with the calculated high ΔG^≠^ values for catalyst detachment in Table 6.

For the reaction with the catalyst systems K_2_SnO_3_ and K_3_PO_4_, LC-ESI-MS orbitrap mass spectrometry measurements were made from samples withdrawn during the reaction at defined time intervals. These mass spectra are shown in Figure 14 and Figure 15.

It can be observed from Figure 14 and Figure 15 that after a short reaction time, the masses of the polymers primarily correspond to the masses present at the end of the reaction. Substantial changes in the molecular weight during the progress of the reaction were not observed, indicating no influence of the decreasing monomer concentration on the degree of polymerisation. Therefore, the ring-opening polymerisation reaction behaves similarly to a free radical polymerisation with strong transfer to monomer. Such polymerisation reactions are primarily defined by initiation, growth, termination, and transfer reactions. The termination itself occurs through combination and transfer reactions. If the transfer step is the predominant step for the termination of the growth of the individual chain, the average degree of polymerisation is given by the ratio of the propagation rate to the transfer rate. If both reactions only depend on catalyst and monomer concentration, the concentrations cancel out, and Pn is given by the propagation rate constant to the transfer rate constant (see Equation (5)).
(5)$$Pn=\frac{M_{polymer}}{M_{Monomer}}=\frac{k_{prop.}}{k_{transfer}}$$

Using LC-ESI-MS orbitrap mass spectrometry, the number average molecular weights of the polymers can be obtained. Since an ethylene oxide unit (EO) is always incorporated into the polymer chain during a propagation step, the mass of CO_2_ can be subtracted from the number average molar mass (M_n_) after determining the CO_2_ content via ^1^H-NMR. These masses can be used to calculate the degree of polymerisation via dividing them by 44 g/mol (M ethylene oxide).

Since the propagation rate is defined by the reaction rate constants k_1_ and k_2_, which describe the propagation with and without CO_2_ incorporation, resulting in the CO_2_ selectivity, these constants can be added to the rate constant of chain growth (k_w_). Using these LC-ESI-MS data and Equation (5), the rate constant for the transfer step can be determined, as shown in Table 7.

Plotting the degree of polymerisation against the catalyst concentration, it is apparent that the degree of polymerisation is constant within the experimental error. Since the catalysts are bases, this agrees with the observations of Rokicki et al. [6]. Furthermore, the calculated barriers of nucleophilic attack by the catalyst anions to the partially formed polymer chains are 25 and 38 kcal/mol for PO_4_^3−^ and SnO_3_^2−^, respectively, supporting chain degradation at high temperatures triggered by an increased rate of back binding and acceleration of the transfer reaction. See Figure 16.

By plotting the rate constants of the propagation and transfer reaction (Figure 17), it can be seen that the rate constant of the transfer reaction is smaller than the growth reaction by a factor of about 6 to 10. This difference makes the reaction possible in the first place but inevitably reduces the degree of polymerisation. This finding explains the variations in the molecular weights with different catalysts already observed in the literature [6,13]. 

## 4. Conclusions

In this work, we have investigated alternative paths of the ring-opening polymerisation of cEC monomer with and without CO_2_ release, originating from the catalyst–monomer interaction in the propagation step. The kinetic measurements, in combination with metadynamics simulations, indicate that the mechanism of chain growth (starting from the catalyst–monomer molecular complex with the catalyst attached to the cleaved cEC monomer) has a higher barrier with a more significant probability for CO_2_ release than the two-step mechanism involving the ROH starter molecule in the initiation step. We examined the insertion of the cEC monomer as a nucleophile into the CAT-CH_2_ molecular complex and the detachment of the catalyst from the chain, followed by the chain transfer. The barriers of chain transfer are high; this finding is in accordance with the increase of the possibility of CO_2_ release throughout the cEC ring cleavage. Our mechanistic investigations confirmed that the cEC monomer can rarely be considered as a nucleophile, but instead predominantly as an electrophilic species, which is in accordance with the anionic ROP of cEC proposed in the literature. These computational findings are supported by the experimental kinetic measurements and the results that the calculated rate constants of the chain transfer and catalyst detachment are almost an order of magnitude higher than the chain propagation and CO_2_ release. The calculated barriers confirm that the nucleophilic attack throughout the second anionic oxygen of the catalyst anion can promote chain transfer and catalyst detachment. This realisation means that with the formation of the second cleaved cEC and the start of a new polymerisation channel, the former generated chain leaves the catalyst. This transfer can lead to polymer chains with specific shorter lengths and keeps the molecular weights constant. The results obtained in this study can help in designing novel catalysts with improved characteristics to produce polymers with higher CO_2_ content. For higher molecular weights of the polymers, catalysts with higher basicity should be applied, with a lower tendency to detach from the growing chain.

## Data Availability

The data will be made available upon request.

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
