# Peer review of "Catalytic Ring-Opening Polymerisation of Cyclic Ethylene Carbonate: Importance of Elementary Steps for Determining Polymer Properties Revealed via DFT-MTD Simulations Validated Using Kinetic Measurements"

_polymers, 2023, doi:10.3390/polym16010136_

Round 1

Reviewer 1 Report

Comments and Suggestions for Authors

Dear Editor

The article entitled ‘The importance of elementary steps induced by ….’ deals with ring-opening polymerization of cyclic ethylene carbonate, both experimentally and by molecular simulation. It is a clean research. I recommend for acceptance after a minor revision as the following:

1- The references are almost old. It is suggested to use recently published articles in the literature review section.

2- The quality of Figures 6, 7 and 17 is not good.

Reviewer 2 Report

Comments and Suggestions for Authors

This work provides an in-depth understanding of the factors influencing the ring opening polymerization (ROP) of the cyclic ethylene carbonate (cEC) monomer. The authors have conducted an extensive computational study of the interaction between the anionic catalyst and the cEC monomer leading to different polymerization pathways with and without CO2 release. They conclude that the barrier for chain growth via alcohol as the nucleophile is lower than the nucleophilic attack of the catalysts on cEC. It was also observed that the nucleophilic attack of cEC monomer on cEC-cat complex has a high free energy barrier indicating that the cEC monomer prefers to act as an electrophile. The chain thus propagates either via the nucleophilic attack (via COO-) of the cEC-cat complex on the cEC monomer, or the insertion of the cEC monomer to the cEC-cat complex. The mechanism of insertion varies with the catalyst used. The results obtained are also supported by experimentally determined kinetics of these reactions.

The use spectrophotometric measurements to study kinetics is a common practice and thus I would not comment on that. However, some clarifications are required before these results can be published:

1) Line 223-225: It is not clear how the authors reach to the conclusion that a longer bond distance performs better in keeping higher CO2 content in the product. At this point in the text, the authors have not begun to discuss or refer to their results in Table 2. Besides, Sn--O bond distance is greater than V--O bond distance, however, the data in Table 2 shows that CO2 content with K3VO4 is greater than that with K2SnO3. This is contrary to the statement made. I request the authors to please provide a detailed explanation for this. Additionally, I suggest the authors to provide the references to the supporting data while making such statements.

2) In section 3.1, the authors conclude that the ring opening via attack of K3PO4 happens in a concerted fashion. However, Figure 4 shows a considerable population of the intermediate state, indicating that the simulation samples this pathway. Additionally, in Figure 6, a short lived intermediate can also be seen. It appears that the reaction can proceed via a concerted pathway but the two-step mechanism is preferred. To further resolve this, it would be helpful if the authors could compare the experimentally obtained rate constants to the rate constants corresponding to the energy of the slow step obtained from MTD simulations.

3) The authors computed the barriers for two pathways shown in Figure 11 using MTD simulations. However, they refer to figure 13 to define the CVs, the two mechanisms shown in Figure 13 are completely different. Thus, the barriers for mechanism 11a and 11b remain unclear. Please comment on this.

4) It is unclear how the authors make the following statement based on the data in Table 6 and Figures 11 and 13.
”Since the barriers of the catalyst’s second oxygen attack are lower than the catalyst detachment (Table 6), the second nucleophilic attack begins faster and can promote the chain transfer/catalyst detachment. This finding means that with the formation of the second cleaved cEC and the start of a new polymerisation channel, the older generated chain leaves the catalyst, leading to polymer chains with specific shorter lengths and constant molecular weights.”

In order to make this statement, the catalyst detachment or chain transfer barrier has to computed from a structure where two of the catalyst oxygen atoms are involved in bonding with cEC. In such a case, if the chain transfer barrier is lower than that for the catalyst’s second oxygen attack, the above statement holds.
I suggest the authors to conduct additional simulations in this respect.

5) Please define EO in line 43.

6) Please correct the typographical errors in the percentages mentioned in line 163-166.

7) The authors might consider making the title of the paper shorter.

Comments on the Quality of English Language

Minor editing of English language required

Reviewer 3 Report

Comments and Suggestions for Authors

The article "The importance of elementary steps induced by catalyst-monomer interactions in ring-opening polymerisation of cyclic ethylene carbonate for determining polyether carbonate polyols characteristics: a DFT-metadynamics study validated by kinetic measurements" is devoted experimental and theoretical studies of cEC ROP, taking into account CO2 elimination during polymerization. The level, results of the study, and quality of presentation generally meet high requirements of the Polymers journal. Excessive detail in description of side processes seems not entirely appropriate, but it’s the author’s choice.

I recommend to accept this work for publication.

Comments and recommendations:

Title and Headings – Please capitalize

Abstract – the catalysts (K3VO4, K3PO4, Na2SnO3) should be specified here

Line 77 – This work is devoted to cEC, the common structure of cyclic carbonate in Scheme 2 seems inappropriate, especially since the absence of CO2 elimination during ROP of trimethylene carbonate

Line 145 – The molecular cluster presented in Scheme 3 should be justified. The statement "six cECs can make the first shell interact with the catalyst" should be deployed with more detailed description of the interactions

Line 161 and below – Suppliers of chemicals and equipment should be provided in accordance with the Instructions for Authors (The name of Company, City, State (if appropriate), Country). Please check and correct

Line 177 and below – cm–1 not cm-1

Line 184 and below – 10 mol% not 10 mol-%

Line 197 – please provide correct number of reference

Line 201 – please replace ΔG# by ΔG

Line 202 – strictly speaking, stage II is a multi-step process with a set of activation berries. What specific reaction do the step II ΔG values relate to?

Line 354 – Please use Polymers template for the formatting of the Tables

Line 397 – please remove the spectral data in Figure 8

Line 400 and throughout the manuscript – 170 °C  not 170°C

Line 614 – bad quality of the Figure 17, please improve

References – the template for the citing of the patents is: Author, A.; Author, B. Title of the Patent. US Patent XXX, DD Month 20XX. Please correct Refs. 4, 5

Comments on the Quality of English Language

Moderate editing.

Author Response

pleas see uploaded file

Round 2

Reviewer 2 Report

Comments and Suggestions for Authors

The authors have adequately addressed the concerns raised. The manuscript can now be considered for publication. No further revisions are necessary.

Comments on the Quality of English Language

Minor editing of English language required